# From Athens to Sparta—37 Years of Spartathlon

**DOI:** 10.3390/ijerph18094914

**Published:** 2021-05-05

**Authors:** Beat Knechtle, Margarida Gomes, Volker Scheer, Robert Gajda, Pantelis Theodoros Nikolaidis, Lee Hill, Thomas Rosemann, Caio Victor Sousa

**Affiliations:** 1Medbase St. Gallen Am Vadianplatz, 9000 St. Gallen, Switzerland; 2Institute of Primary Care, University Hospital Zurich, 8091 Zurich, Switzerland; Thomas.rosemann@usz.ch; 3Laboratory of Physical Activity and Health, Polytechnic Institute of Beja, 7800-295 Beja, Portugal; gomesmargarida0@gmail.com; 4Ultra Sports Science Foundation, 69310 Pierre-Bénite, France; volkerscheer@yahoo.com; 5Center for Sports Cardiology, Gajda-Med Medical Center in Pułtusk, 06-100 Pułtusk, Poland; gajda@gajdamed.pl; 6School of Health and Caring Sciences, University of West Attica, 12243 Athens, Greece; pademil@hotmail.com; 7Division of Gastroenterology and Nutrition, Department of Pediatrics, McMaster University, 1280 Main St W, Hamilton, ON L8S 4L8, Canada; hilll14@mcmaster.ca; 8Bouve College of Health Sciences, Northeastern University, Boston, MA 02115, USA; cvsousa89@gmail.com

**Keywords:** endurance, performance, ultra-marathon, athlete, female, male

## Abstract

(1) Background: Recent studies analyzed the participation and performance trends of historic races such as the oldest ultra-marathon (Comrades) or the oldest 100-km ultra-marathon (Biel). One of the toughest and historic ultra-marathons in the world is the ‘Spartathlon’ (246-km ultra-marathon from Athens to Sparta). The present study aimed to analyze the trends in participation and performance of this race. (2) Methods: Different general linear models were applied as follows: the first model was a two-way ANOVA (Decade × Sex), with separate models for all participants and for only the top five finishers in each race; the second model was a two-way ANOVA (Age Group × Sex); the third model was a two-way ANOVA (Nationality × Sex). (3) Results: Between 1982 and 2019, 3504 ultra-marathoners (3097 men and 407 women) officially finished the Spartathlon at least once. Athletes from Japan were the majority with 737 participants, followed by far by runners from Germany (*n* = 393), Greece (*n* = 326), and France (*n* = 274). The nations with the highest numbers of athletes amongst the top five performers were Japan (*n* = 71), followed by Germany (*n* = 59), and Great Britain (*n* = 31). Runners from the USA were the fastest in men, and runners from Great Britain were the fastest in women. Female and male runners improved performance across the decades. The annual five fastest women and men improved their performance over time. Runners achieved their best performance earlier in life (20–29 and 30–39 years) than female runners (30–39 and 40–49 years). Runners in age group 30–39 years were the fastest for all nationalities, except for Greece. (4) Conclusions: Successful finishers in the Spartathlon improved performance in the last four decades and male runners achieved their best performance ~10 years earlier in life than female runners.

## 1. Introduction

Endurance running events are some of the most popular sports worldwide. Beginning in the 1980s [1], the number of ultra-marathon competitions and participants worldwide has grown exponentially [2]. Ultra-marathons are either held as distance-limited runs held in km or miles, or as a time-limited event occurring within a set number of hours or days [3]. Ultra-marathons are longer than the traditional marathon distance of 42.195 km (26.2 miles) [4] or have a minimum time of six hours [5]. Ultra-marathons covering 50 km, 100 km, 50 miles, and 100 miles are most frequently performed ultra-marathons held as distance-limited events [6].

One of the famous ultra-marathons in the world, the ‘Spartathlon’, covers the distance between Athens and Sparta (Greece), with a total distance of 246 km [7] and has been held in Greece since 1983. Spartathlon has been held annually since 1983; in 1982, five Royal Air Force officers attempted to run the route that Pheidippides had run about 500 years earlier [8].

It is one of the most difficult ultra-marathons in the world due to is unique history. The historical route from Athens to Sparta must be covered in a time limit of 36 h. The father of the Spartathlon is the Greek messenger Pheidippides, who was—according to the reports of Herodotus—sent in 490 BCE to Sparta by the Athenians during the Persian Wars to ask the Spartans for help in the upcoming Battle of Marathon. The messenger set out on the 246-km route in the morning and arrived the next day in the evening [9].

The Spartathlon is a historic ultra-marathon taking place annually in September. The course covers rough tracks and muddy paths, crosses vineyards and olive groves and climbs steep hillsides. In the night, it takes the runners on the Mount Parthenion (Παρθένιο—Parthenio) with an altitude of 1215 m above sea level. Pan appeared to Philippides on Mt. Parthenion above Tegea before the battle of Marathon in 490 BCE. Temperatures can be as low as 4 °C on Mount Parthenion (Spartathlon 2021).

Acute physiological and pathophysiological responses to Spartathlon have been examined regarding several aspects, including bone metabolism [10], heart function [11], incidence of hyponatremia [12], and circulating stress-related proteins [13]. Only the world’s best ultra-marathoners are able to finish the event or to win Spartathlon [14,15].

However, limited information exists on performance and participation trends in this race regarding age group runners [16,17], and no study has analyzed the trends of the last decade. Therefore, an updated analysis of the recent trends in elite participation and performance is needed.

In these ultra-endurance events, non-elite (recreational) athletes, also called age group athletes, have drawn particular attention, especially with regard to their running speed [2]. When analyzing running speed by age group, the fastest athletes were in the age group of 30–39 years in men and 40–49 years in women [18]. However, sex differences in performance have been changing over time. Men below the age of 49 years reached a peak of participation in the 1980s and have decreased since [2]. However, when only the fastest runners are examined, women tend to peak at a younger age than men [19,20]. This dynamic is probably due to an increasing number of master athletes, especially women [21,22].

From a practical perspective, knowledge of performance and participation trends might aid strength and conditioning coaches in developing sex-, age-, and performance-tailored training programs [21]. For instance, information about the age of peak performance, i.e., when it is expected that an athlete reaches her peak, would be crucial when setting long-term training goals.

Therefore, the present observational, retrospective, and cross-sectional study aimed to investigate trends in performance and participation in the Spartathlon ultra-marathon for the four decades from 1982 to 2019, focusing on age groups, gender, and running speed. We hypothesized that the number of finishers would increase over time. Additionally, we hypothesized that the increased number of participants would increase the average race time across calendar years, but that the top finishers would be faster due to advancements in nutrition (i.e., diets and supplements), training methods, and equipment [22].

## 2. Materials and Methods

### 2.1. Ethics Approval

This study was approved by the Institutional Review Board of Kanton St. Gallen, Switzerland, with a waiver of the requirement for informed consent of the participants as the study involved the analysis of publicly available data (EKSG 1 October 2010).

### 2.2. The Race

The Spartathlon is an ultra-marathon covering a distance of 246 km and must be completed within 36 h. Entry requirements for Spartathlon are: in the three years prior to the race, athletes must register a completion of an ultra-marathon such as 120 km (men) or 110 km (women) in a 12-h race, finish a 100-mile race in 21:00 h:min (men) or 22 h:min (women), cover 180 km (men) or 170 km (women) in a 24-h race, finish ‘Western States 100-Mile Endurance Run’ within 24:00 h:min (men) or 25:00 h:min (women), finish ‘Badwater’ within 39:00 h:min (men) or 40:00 h:min (women) among other criteria [23]. During the race, the athletes have to pass a total of 74 check points within a time limit [24]. When an athlete is not able to pass the checkpoint within the time limit, they are required to withdraw from the race. The strict criteria for qualification and the strict time limits during the race lead a collection of the best ultra-marathoners in the world to enter this race.

### 2.3. Data

Data were obtained from the official Spartathlon race website [25]. For each athlete, sex, age, country of origin, and race time (h:min:s) were recorded.

### 2.4. Statistical Analysis

The dependent variable was average running speed (km/h). Descriptive statistics are presented as absolute values (participation outcomes) and means and standard deviations (performance outcomes). Different general linear models (GLM) were applied: the first model was a two-way ANOVA (Decade × Sex), with separate models for all participants and for only the top five finishers in each race; the second model was a two-way ANOVA (Age Group × Sex); the third model was a two-way ANOVA (Nationality × Sex). The factor ‘decade’ included four groups of calendar years: 1982–1989, 1990–1999, 2000–2009, and 2010–2019. The factor ‘nationality’ considered the nations with most participants across calendar years; all other nations were grouped as ‘other’. The fourth general linear model was also a two-way ANOVA (Nationality × Age Group).

Non-linear trend lines were calculated using a quadratic function for performance across calendar years for all participants, top five finishers, and winners. All statistical analyses were carried out using the Statistical Software for the Social Sciences (IBM^®^ SPSS v.25, Chicago, IL, USA).

## 3. Results

Between 1982 and 2019, 3504 ultra-marathoners (3097 men and 407 women) officially finished the Spartathlon. Table 1 summarizes the number of participants in each subgroup (i.e., by decade, age group, and nationality) of the ANOVA analyses.

Women represented 11.6% of the total finishers (Figure 1A). The age group with the highest number of participants for both men and women was the age group 40–49 years, followed by the age groups 30–39 and 50–54 (Figure 1B). Athletes from Japan (JPN) were the majority with 737 participants, followed by far by runners from Germany (GER; *n* = 393), Greece (GRE; *n* = 326), and France (FRA; *n* = 274). The nations with the highest numbers of athletes amongst the top five performers were Japan (JPN; *n* = 71), followed by Germany (GER; *n* = 59), and Great Britain (GBR; *n* = 31) (Figure 1C).

In the first GLM (Decade × Sex) we observed a significant effect for sex (*F* = 10.713; *p* = 0.014) but not for decade (*F* = 1.165; *p* = 0.452). Pairwise comparisons showed that men were faster in two decades, 1982–1989 and 2000–2009 (Figure 2A). Considering only the top five performers, the same model identified a significant effect for both sex (*F* = 316.67; *p* < 0.001) and decade (*F* = 24.033; *p* = 0.013). Pairwise comparisons showed that men were faster in all decades, and both men and women were faster in 2000–2009 and 2010–2019 (Figure 2B). In the second GLM (Age Group × Sex), we observed significant effects for age group (*F* = 5.225; *p* = 0.027) and sex (*F* = 6.813; *p* = 0.016).

Pairwise comparisons showed that men were fastest in age groups 20–29 and 30–39 years, whereas women were fastest in age groups 30–39 and 40–49 years (Figure 2C). Performance trends with all athletes showed a neutral trend for both men and women (Figure 2D), but when considering the annual top five finishers (Figure 2E) or winners (Figure 2F), the trend showed a positive performance increase for both men and women.

The third general linear model (Nationality × Sex) showed a significant effect for nationality (*F* = 4.653; *p* = 0.042) but not for sex (*F* = 2.400; *p* = 0.141). For men, pairwise comparisons showed that athletes from Japan (JPN) were significantly (*p* < 0.05) slower than athletes from Germany (GER), Great Britain (GBR), the United States of America (USA), and others. Runners from Germany (GER) were significantly (*p* < 0.05) faster than runners from Japan (JPN), Greece (GRE), and France (FRA). Finishers from the United States of America (USA) were significantly (*p* < 0.05) faster than finishers from Japan (JPN), Greece (GRE), and France (FRA). For women, pairwise comparisons showed that runners from Japan (JPN) were significantly (*p* < 0.05) slower than runners from Great Britain (GBR), the United States of America (USA), and others. Finishers from France (FRA) were significantly (*p* < 0.05) slower than finishers from Great Britain (GBR) and the United States of America (USA) (Figure 3A).

The fourth general linear model (Nationality × Age Group) showed a significant effect for both nationality (*F* = 2.460; *p* = 0.027) and age group (*F* = 10.563; *p* < 0.001). The athletes in age group 30–39 years were the fastest for all nationalities, except for Greece (Figure 3B).

## 4. Discussion

This study intended to investigate participation and performance trends in the Spartathlon regarding sex, age, and nationality. The main findings were (i) most of the finishers originated from Japan, but male American and female British runners were the fastest; (ii) female and male runners improved performance across decades, where the annual top five and annual winners improved; and (iii) male runners achieved their best performance earlier in life (20–29 and 30–39 years) than female runners (30–39 and 40–49 years).

The most important finding was that the fastest runners (i.e., annual winners and annual top five) improved their performance over the preceding four decades. Considering all women and men, performance was observed to be stable over the years, but when considered by decades, both women and men were faster in the last two decades compared with the first two decades of the race. There was an improvement in performance for the fastest runners in the Comrades Marathon (South Africa) during the century 1921–2019; whereas overall women and men maintained a stable running speed, an improvement in performance for the annual top five women and men was found [26]. Both the Comrades Marathon and the Spartathlon have strict participant selection criteria and strict cut-off times during the race. The fastest runners’ improvement in performance might be attributed to advances in sport science and technology during the period we examined [22].

In contrast, in the ‘100-km Lauf Biel’ in Switzerland, the oldest 100-km ultra-marathon in the world, held since 1956, both overall women and men and both the women and men who won were not able to improve race times after ~1985 [2]. This can be explained by the number of participants and the selection criteria for participants. In the 100-km Lauf Biel, the number of male participants reached a peak in ~1985, and a decline in participation occurred thereafter [2]. In contrast to Comrades Marathon and Spartathlon, interested participants in the 100-km Lauf Biel need not fulfil any selection criteria and have a time limit of 21 h. In the Western States 100-Mile Endurance Run, since its inception in 1974, women improved between 1986 and 2007, whereas performances among the men did not improve over this time span [6]. When performance trends in 100-mile (161-km) ultra-marathon races held in North America between 1977 and 2008 were analyzed, female race times improved relative to male race times throughout the 1980s but then remained stable. Furthermore, the fastest runners’ race times showed no improvements over years [27]. An increase in participation will not necessarily lead to an improvement in performance. This has also been shown when all 100-km ultra-marathons held since 1960 were analyzed [28]. There was a substantial increase in both the number of running events and in the number of participants, where performance started to decline after the year 2000 for runners younger than 70 years, with an improvement only in runners older than 70 years [28]. Therefore, future improvement in the elite ultra-marathoners’ performance can be expected only in races with strict criteria for selection and strict cut-off times during the race.

A further important finding was that men achieved their best performance in Spartathlon about 10 years earlier in life (20–29 and 30–39 years) than women (30–39 and 40–49 years). In marathon running, the age of the best performance has been reported to peak earlier [29,30] or later [31] in life in women compared with men, depending upon the analyzed races. However, the differences were only about 2 years [29] to 5 years [30], not about 10 years.

In ultra-marathon running, women achieved the best race times in a 50-km ultra-marathon later in life compared with men [32]. In 100-km ultra-marathon running, however, the age of peak performance was younger in women than in men [33]. There are, however, no data about the sex difference in the age of peak performance for ultra-marathon races longer than the 100-km race distance. It is well-known that the age of peak ultra-marathon performance increases with the increasing length of a race. In time-limited ultra-marathons held from 6 h to 10 days during 1975–2013, the age of the best ultra-marathon performance increased with increasing race duration [34].

Regarding participation, the observation that the largest number of finishers were Japanese was in agreement with the existing literature in ultra-marathon races [35]. In particular, an analysis of 150,710 athletes who finished a 100-km ultra-marathon between 1959 and 2016, Japanese was the first nationality in participation [35]. The high rate of Greek finishers confirmed the trend of high participation of local athletes in the race, shown in the Comrades [36]. The high rates of Japanese participation implied that these runners were more ‘recreational’ or less ‘selective’ than runners from other nationalities, which, in turn might explain why they had the slowest race time. Spartathlon has specific demands related to geographical (ascents and descents through the mountains of the Peloponnese peninsula) and environmental conditions (e.g., hot weather); thus, the findings should be generalized with caution to ultra-marathons with different characteristics. The study’s strength was that it added novel information about trends in participation and performance during the last decade in the abovementioned race. Coaches and trainers might use this information to set sex-, age-, and nationality-tailored training goals.

A last important finding was that runners in the age group 30–39 years were the fastest for all nationalities except local runners from Greece. This finding suggests that only the best ultra-marathoners in the world compete in this race in which local runners were not able to keep up with the world’s leading ultra-marathoners.

## 5. Conclusions

In the 246-km Spartathlon, the fastest runners (i.e., the annual winners and the annual top five) improved their performance over the last four decades. Considering all women and men, running performance was found to be stable over the years, but when considered by decades, both women and men were faster in the last two decades compared with the first two decades of the race. This improvement is most likely due to the participant selection criteria and the strict cut-off times during the race. Men achieved their best performance in Spartathlon about 10 years earlier in life (20–29 and 30–39 years) than women (30–39 and 40–49 years). Future studies need to investigate the age of peak ultra-marathon performance and any sex differences in performance by age in ultra-marathon races longer than 200 km.

## Figures and Tables

**Figure 1 ijerph-18-04914-f001:**
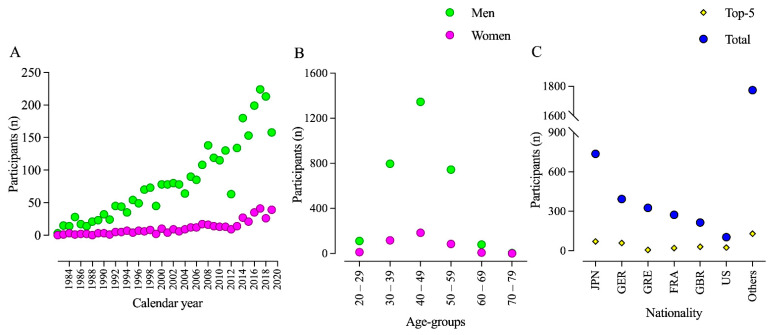
Participation trends in the Spartathlon by calendar year (**A**), age groups (**B**), and nationality (**C**).

**Figure 2 ijerph-18-04914-f002:**
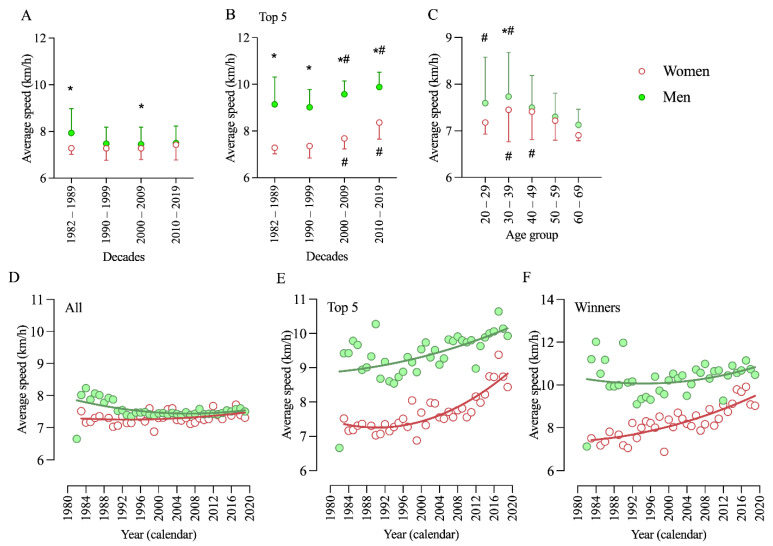
Performance trends in the Spartathlon across decades and age groups. *: statistically significant difference between men and women; #: statistically significant difference between decade or age group (*p* < 0.05). (**A**) Spartathlon performance of men and women by decade; (**B**) Spartathlon performance of men and women top-5 finishers by decade; (**C**) Spartathlon performance of men and women by age group; (**D**) nonlinear regression of men and women performance by calendar year; (**E**) nonlinear regression of performance by calendar year of men and women top-5 finishers; (**F**) nonlinear regression of performance by calendar year of the winners in each race.

**Figure 3 ijerph-18-04914-f003:**
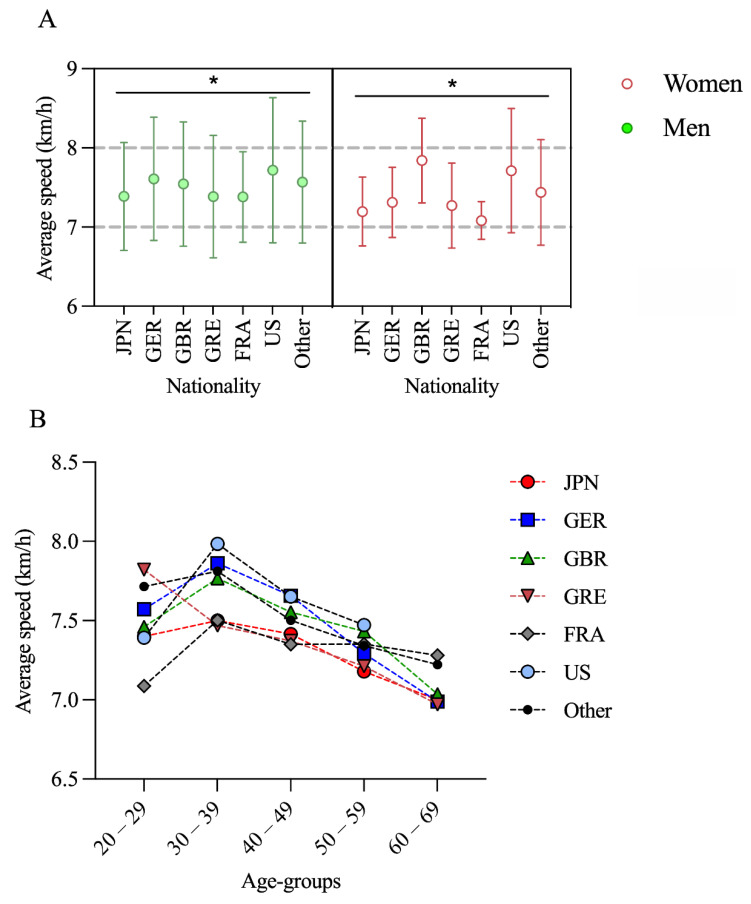
(**A**) Performance of men and women from different nationalities in the Spartathlon; (**B**) performance by nationality and age groups; *: statistically significant nationality effect (*p* < 0.05).

**Table 1 ijerph-18-04914-t001:** Number of participants in each subgroup of the ANOVA analyses. Data presented as *n*.

Figure 2A	Men	Women
1982–1989	135	12
1990–1999	471	48
2000–2009	918	109
2010–2019	1569	238
Figure 2B		
1982–1989	37	12
1990–1999	49	40
2000–2009	50	48
2010–2019	50	50
Figure 2C		
20–29	110	12
30–39	797	117
40–49	1344	184
50–59	745	84
60–69	79	7
Figure 3		
Japan	611	126
Germany	335	62
Great Britain	198	17
Greece	321	11
France	262	12
USA	74	29
Other	1292	150

## Data Availability

All data are available from Spartathlon database [25].

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
