# Peer review of "From Athens to Sparta—37 Years of Spartathlon"

_ijerph, 2021, doi:10.3390/ijerph18094914_

Round 1
Reviewer 1 Report
Thank you for the opportunity to review this manuscript. The article contains a statistical analysis of performance and the participation trends in “Spartathlon” race regarding age group and runners nationality.
Introduction
I would suggest developing the introduction a bit. What makes this ultra-marathon so special? What is the sum of height difference?; whether it is always performed in the same time span or is it a moving event?; what is the average temperature during the run over the years?
Second paragraph:
A world-famous ultra-marathon, the ‘Spartathlon’ – I would rather said “one of the famous…’
Spartathlon is held annually since 1983, in 1982, five Royal Air Force officers attempted to run the route that Pheidippides had run about 500 years earlier.
Seventh paragraph: Sport sciences - too laconic, too general. I suggest the authors expand their thoughts a bit: the diet and supplementation, training methods, training loads, etc.
Discussion:
Why nationality is important and why it was analyzed. Does this information add anything, apart from pure statistics?
Author Response
Reviewer 1
Thank you for the opportunity to review this manuscript. The article contains a statistical analysis of performance and the participation trends in “Spartathlon” race regarding age group and runners nationality.
Answer: We appreciate the reviewer’s input and suggestions in our manuscript.
Introduction
I would suggest developing the introduction a bit. What makes this ultra-marathon so special? What is the sum of height difference?; whether it is always performed in the same time span or is it a moving event?; what is the average temperature during the run over the years?
Answer: We agree with the expert reviewer and took some basic information from the website of the race ‘The ‘Spartathlon’ is a historic ultra-marathon taking place annually in September. The course covers rough tracks and muddy paths, crosses vineyards and olive groves and climbs steep hillsides. In the night, it takes the runners on the Mount Parthenion (Παρθένιο – Parthenio) with an altitude of 1,215 meters above sea level. Pan appeared to Philippides on Mt. Parthenion above Tegea, before the battle of Marathon in 490 BCE. Temperatures can be as low as 4 °C on Mount Parthenion (www.spartathlon.gr).’
Second paragraph:
A world-famous ultra-marathon, the ‘Spartathlon’ – I would rather said “one of the famous…’
Answer: We agree with the expert reviewer and changed to ‘One of the famous ultra-marathons in the world, the ‘Spartathlon’, covers the distance between Athens and Sparta (Greece), with a total distance of 246 km (www.spartathlon.gr) and has been held in Greece since 1982.
Spartathlon is held annually since 1983, in 1982, five Royal Air Force officers attempted to run the route that Pheidippides had run about 500 years earlier.
Answer: We agree with the expert reviewer and changed to ‘Spartathlon is held annually since 1983, in 1982, five Royal Air Force officers attempted to run the route that Pheidippides had run about 500 years earlier (www.spartathlon.gr/en/the-spartathlon-race-en/historical-information-en.html)’
Seventh paragraph: Sport sciences - too laconic, too general. I suggest the authors expand their thoughts a bit: the diet and supplementation, training methods, training loads, etc.
Answer: We agree with the expert reviewer and changed to ‘The increased number of participants would increase the average race time across calendar years, but the top finishers would be faster due to advancements in nutrition (i.e., diets and supplements), training methods, and equipment’.
Discussion:
Why nationality is important and why it was analyzed. Does this information add anything, apart from pure statistics?
Answer: We performed further analyses and added a fourth model for nationality x age group. We found that runners in age group 30-39 years were the fastest for all nationalities except for Greece. We added this aspect at the end of the discussion with ‘A last important finding was that runners in the age group 30-39 years were the fastest for all nationalities, except local runners from Greece. This finding suggests that only the best ultra-marathoners in the world compete in this race where local runners were not able to keep up with the world leading ultra-marathoners.’ We also add this aspect in the abstract.
Reviewer 2 Report
Thank you very much by invitation to evaluate the manuscript: “From Athens to Sparta—37 years of Spartathlon”. It was a greatest opportunity to learn.
Regarding my evaluation report. The present study aimed to investigate trends in performance and participation in the Spartathlon ultra-marathon for four decades (1982–2019), focusing on age groups, gender, and running speed. The authors achieved the study purposes, the writing in English language is very adequate, the study have all scientific criteria to be a very good evidence in Ultra-marathon studies scope. Considering this, my suggestions are only details in methods of study, that could better clearly for understanding of readers.
1. The study needs assuming a research Design, for example: is it a descriptive, a longitudinal, a cross sectional, a documental, a follow-up, a data base review?
2. The number of in participants in each group of comparison could be presented in figures.
3. How the authors minimized the errors of ANOVA, considering the different number of participants in each subgroup data analyses?
Congratulations by this beautiful research.
Author Response
Reviewer 2
Thank you very much by invitation to evaluate the manuscript: “From Athens to Sparta—37 years of Spartathlon”. It was a greatest opportunity to learn.
Regarding my evaluation report. The present study aimed to investigate trends in performance and participation in the Spartathlon ultra-marathon for four decades (1982–2019), focusing on age groups, gender, and running speed. The authors achieved the study purposes, the writing in English language is very adequate, the study have all scientific criteria to be a very good evidence in Ultra-marathon studies scope. Considering this, my suggestions are only details in methods of study, that could better clearly for understanding of readers.
- The study needs assuming a research Design, for example: is it a descriptive, a longitudinal, a cross sectional, a documental, a follow-up, a data base review?
Answer: We agree with the expert reviewer and changed to ‘Therefore, the present observational, retrospective and cross-sectional study aimed to investigate trends in performance and participation in the Spartathlon ultra-marathon for four decades (1982–2019), focusing on age groups, gender, and running speed’.
- The number of in participants in each group of comparison could be presented in figures.
Answer: We agree that the number of participants in each sub-group is important. But adding this to the figures would make it confusing (too much information). Thus, we created a Table with all this data that is now included as supplementary material. - How the authors minimized the errors of ANOVA, considering the different number of participants in each subgroup data analyses?
Answer: We agree with the expert reviewer. The general linear model we applied considered the estimated marginal means (or unweighted means) of each subgroup, thus minimizing to potential confounder of unequal sample sizes.
Congratulations by this beautiful research.
Reviewer 3 Report
This paper is about the performance, and the athletes of ultra races. It includes races of the Spartathlon (and ultra race including 246 km) since 1982 to 2019. In total 3504 runners were included. The paper makes an analysis of the origin, age and performance.
The presented data is interesting. But the authors fail to present much additional information based on the statistical analyses. With that it seems to be only an analysis of the finisher of this specific race.
The discussion is nice but not necessary based on the performed analysis.
Author Response
Reviewer 3
This paper is about the performance, and the athletes of ultra races. It includes races of the Spartathlon (and ultra race including 246 km) since 1982 to 2019. In total 3504 runners were included. The paper makes an analysis of the origin, age and performance.
The presented data is interesting. But the authors fail to present much additional information based on the statistical analyses. With that it seems to be only an analysis of the finisher of this specific race.
Answer: We performed further analyses and added a fourth model for nationality x age group. We found that runners in age group 30-39 years were the fastest for all nationalities except for Greece. We added this aspect at the end of the discussion with ‘A last important finding was that runners in the age group 30-39 years were the fastest for all nationalities, except local runners from Greece. This finding suggests that only the best ultra-marathoners in the world compete in this race where local runners were not able to keep up with the world leading ultra-marathoners.’ We also add this aspect in the abstract. We also add a table with the numbers of participants for each subgroup. Feel free to make specific suggestions what we should change.
The discussion is nice but not necessary based on the performed analysis.
Answer: We think that we correctly discuss the findings. We present the main findings and discuss them stepwise. Feel free to make specific suggestions what we should change.
Round 2
Reviewer 3 Report
Thank you for the revision of the manuscript.